microsystems/mechanical engineering/biomedical engineering

EWOD, radioisotope manipulation, microdroplet, digital microfluidics

**Author for correspondence:**
Katsuo Mogi
e-mail: mogi.k@aist.go.jp

This article has been edited by the Royal Society of Chemistry, including the commissioning, peer review process and editorial aspects up to the point of acceptance.

# Automatic radioisotope manipulation for small amount of nuclear medicine using an EWOD device with a dimple structure

Katsuo Mogi[1], Hiroyuki Kimura[2], Yuto Kondo[2], Tomoya Inoue[3], Shungo Adachi[1] and Tohru Natsume[1]

[1]Cellular and Molecular Biotechnology Research Institute (CMB), National Institute of Advanced Industrial Science and Technology (AIST), Tokyo 135-0064, Japan
[2]Department of Analytical and Bioinorganic Chemistry, Kyoto Pharmaceutical University, Kyoto 607-8412, Japan
[3]Research Centre for Ubiquitous MEMS and Micro Engineering, National Institute of Advanced Industrial Science and Technology (AIST), Ibaraki 305-8564, Japan

KM, 0000-0002-6676-6995

Delicate animal experiments and microdose clinical trials using short-lived radioisotopes require rapid preparation with high accuracy and careful attention to safety within a limited timeframe. We have developed an open-style electrowetting on dielectric (EWOD) device containing dimple structures for the rapid preparation of radiolabelled reagents. The device was demonstrated by automatic preparation of a technetium diethylenetriamine pentaacetate ($^{99m}$Tc-DTPA) with high chelation efficiency (99.7 ± 0.13%). Additionally, we demonstrated the single-photon emission computed tomography/computed tomography imaging of mouse kidney using the $^{99m}$Tc-DTPA prepared with the EWOD device. The obtained organ tomographic images were sufficient for the evaluation of mouse models for specific diseases. These results indicate that manual radiolabelling for a small amount of nuclear medicine can be replaced by a process using the proposed EWOD device as a human error reduction technique.

## 1. Introduction

Imaging techniques such as positron emission tomography (PET) and single-photon emission computed tomography (SPECT) play an important role in diagnosing several diseases such as cancer, vasculitis, cardiac sarcoidosis, epilepsy and dementia [1,2].

Additionally, those techniques are useful means for pharmacokinetic monitoring of drug candidate substances such as low-molecular compounds, peptides and antibodies. A non-clinical process of new drug development requires the accumulation of large amounts of pharmacokinetic data by experimenting with the administration of drug candidate substances to animals prior to a first-in-human study. Furthermore, the first-in-human study requires a microdose clinical trial to investigate its efficacy and side effects of those substances. These delicate animal experiments and microdose clinical trials require sensitive imaging technique such as PET and SPECT, which are minimally invasive and capable of highly accurate measurement. Many experiments using these imaging techniques require high-mix low-volume production techniques for radiolabelling due to the low injection volume of the reagents. On the other hand, the current automatic preparation device installed in a hot cell is unsuitable for preparing a small amount of reagent because the device has a large dead volume. Moreover, complicated preparation and customizing, such as attachment and detachment of expensive disposable parts and checking and cleaning of contamination spots, is required for each process. Thus, the running cost is high and it is not possible to easily prepare small amounts of reagents. Therefore, in order to carry out labelling treatment without losing rare and expensive new drug candidate substances, it is necessary to manually prepare reagents within a limited timeframe [3–5].

To address these challenges, microfluidic devices based on microTAS technology [6] can automatically process small volumes of reagents, minimize the processing time constraints and reagent costs [7,8]. The use of a microfluidic device for medicinal preparation allows accurate processing without human pipetting error, while radioisotope scattering and contamination can be suppressed within the enclosed microchannel space [9–11]. Moreover, high-speed reaction using high-concentration trace reagents eliminates the need for filtration of unreacted substances and eliminates concentration treatment by solid-phase extraction or evaporation. In cases where additional processing such as pH adjustment, dilution with physiological saline or concentration adjustment is required, the microchannel design can be changed accordingly. However, it is difficult to remove the solution in the microchannel since the reagent in the enclosed space can only be accessed from the inlet and outlet ports. In addition, since the processed solution flows continuously through the microchannel from inlet to outlet, most of the solution remains in the microchannel as a dead volume. Likewise, complicated liquid delivery mechanisms, for example, a syringe pump on the outside of the microchannel, can also contribute to the dead volume [12–14].

On the other hand, open-style electrowetting on dielectric (EWOD) devices based on digital microfluidics [15–17] can be expected to solve the problems associated with closed-style microchannels. The open-style EWOD is used to manipulate discontinuous microdroplets without complicated liquid delivery mechanisms. In addition, since the input and output of the solution is simply putting and picking up the droplets on the substrate of the EWOD device, it is also easy to interrupt the process without dead volume [18].

However, considerable attention is required to prevent unwanted droplet movements during experimental procedures caused by various experimental conditions (e.g. substrate tilt and distortion) which is a major issue encountered with open-style EWOD substrates. As a solution to the challenge, we previously developed an open-style EWOD device for microdroplet manipulation [19]. Our EWOD device comprised a thin-film substrate incorporating a 'dimple structure' to avoid erroneous movements caused by the experimental conditions.

In the present study, we have developed a new device based on our previous design that can rapidly and precisely perform radiolabelling processes for the preparation of a small amount of reagent within a limited timeframe. The usability of this device was initially evaluated by a simple chelation of technetium ($^{99m}$Tc) with diethylenetriaminepentaacetic acid (DTPA) as a radiolabelling model. We also demonstrated that the technetium diethylenetriamine pentaacetate ($^{99m}$Tc-DTPA) prepared with this device can be used for SPECT/computed tomography (CT) imaging of mice.

# 2. Material and methods

## 2.1. Model experiment for radiolabelling

As a model experiment for radiolabelling, a $^{99m}$Tc-DTPA injection [20] was prepared using our new EWOD device. This injection is commonly used as a scintigraphic reagent for the diagnosis of renal function and is prepared by mixing a solution of DTPA and sodium (figure 1). In this study, 10 µl DTPA saline solution (0.05 mg µl$^{-1}$; Techne(r)DTPA Kit; Fujifilm RI Pharma, Co., Ltd, Tokyo, Japan)

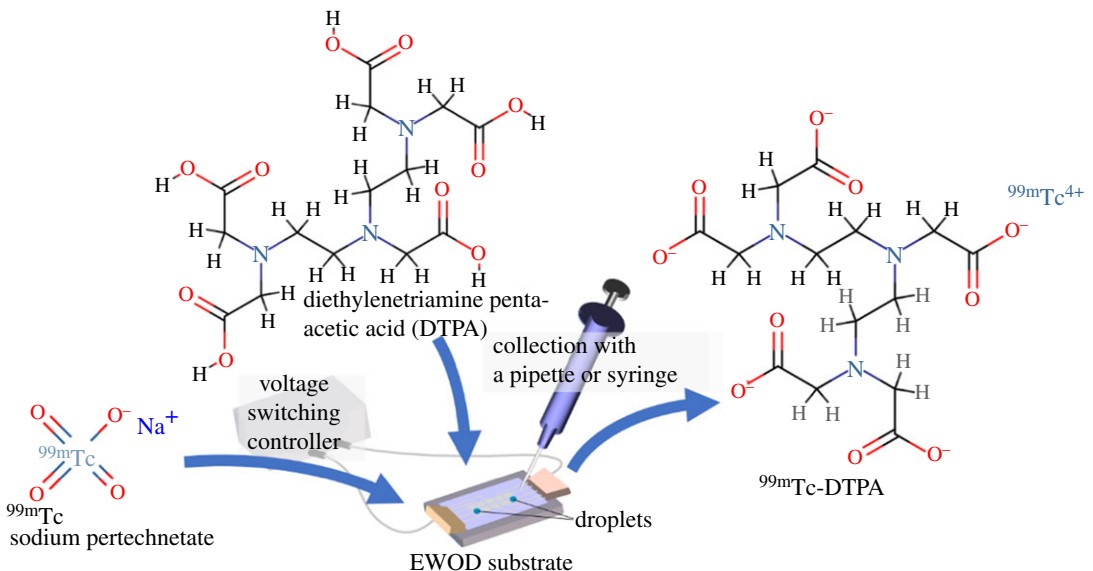

**Figure 1.** Schematic of the chelation of $^{99m}$Tc-DTPA using the proposed EWOD device. $^{99m}$Tc sodium pertechnetate and DTPA droplets are automatically combined and mixed on the EWOD substrate for preparation. All structural formulae are drawn with reference to PubChem, USA (https://pubchem.ncbi.nlm.nih.gov/).

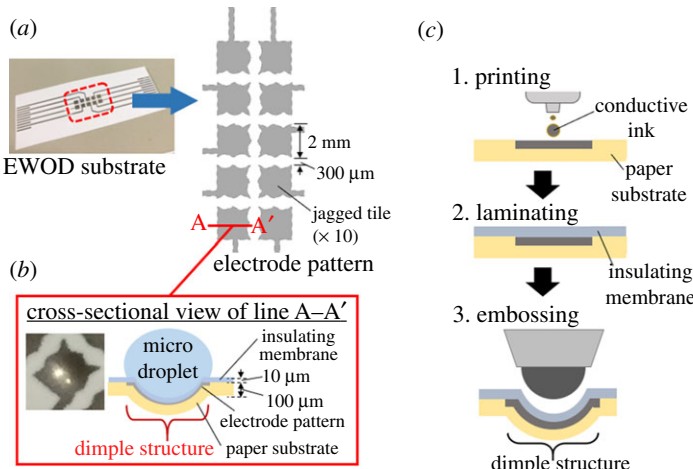

**Figure 2.** Schematics of the EWOD substrate structure. (*a*) Outline and electrode pattern of the substrate; 2 mm square jagged tiles are aligned with 300 μm spacing. (*b*) Cross-sectional view of line A–A′. The thin-film substrate is composed of an insulating membrane and the paper on which the electrodes are printed. A dimple structure with a spherical-cap shape is formed on each tile. (*c*) The fabrication process of a dimple structure on the EWOD substrate.

and 10 μl sodium pertechnetate ($^{99m}$TcO$_4^-$; Tc-10 M; Nihon Medi-Physics Co., Ltd, Tokyo, Japan) were automatically mixed using the EWOD device. The radiation dose was measured with an IGC-8 ALOKA Curiemeter (Hitachi Healthcare, Tokyo, Japan).

## 2.2. EWOD device with dimple structure

We developed an automatic radioisotope manipulation system based on an EWOD device that can use expensive drug candidate substances without waste. The system consisted of an EWOD substrate of paper, a voltage switching controller and a laptop. Droplets on the EWOD board were manipulated by voltage switching with the palm-sized controller. The voltage switching was controlled using Python script on the laptop.

The EWOD substrate used in the experiment was composed of a 300 μm thick paper and a 10 μm thick insulating membrane (figure 2*a*) [19]. On the paper, an electrode pattern consisting of 10 jagged tiles (dimensions, 2 × 2 mm; inter-tile spacing, 300 μm) and conductive paths were printed with

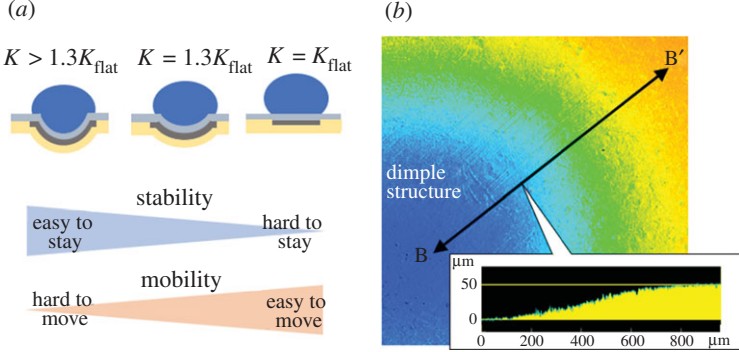

**Figure 3.** Design index and optimum shape of the dimple structure. (*a*) The dimple structure is designed using the coefficient (adhesion energy: *K*) obtained by Wolfram's empirical formula as an index. The $K_{flat}$ is the coefficient of a flat substrate. The optimum shape of dimple structure which has the best balance between mobility and stability is achieved with $K = 1.3K_{flat}$. (*b*) Contour plot of a dimple structure. Point B shows the centre of the dimple structure, and point B' shows a flat position. The depth of the dimple structure is $52.9 \pm 2.5$ μm, which is the average and the standard deviation of five dimples.

conductive ink using an inkjet printer (figure 2*b*) [19,21]. The dimple structure on each tile was formed by embossing the paper substrate laminated with an insulating membrane (figure 2*c*) [19].

The dimple structure can increase the stability of the droplets on each tile and avoid the perturbation that was a major challenge for EWOD. On the other hand, it was necessary to design the optimum shape of the structure because excessive stability causes droplet sticking. We have already shown a method for estimating the optimum dimple structure from the coefficient (the adhesion energy: *K*) obtained by Wolfram's experimental formula (2.1) [22].

$$K = \frac{mg \sin \alpha}{2\pi r}, \tag{2.1}$$

where *m* is the droplet mass, *g* is the acceleration due to gravity, *α* is the critical tilting angle and *r* is the droplet contact radius. When the *K* is approximately 1.3 times that of a flat substrate, the dimples have the best balance between mobility and stability (figure 3*a*). Figure 3*b* shows the optimum dimple structure with a depth of $52.9 \pm 2.5$ μm used in this study.

Since the droplets were stable in the dimple structure of the water-repellent-coated substrate, it was possible to suppress droplet loss from substrate tilt and distortion. Microdroplet manipulation from 5 to 30 μl was possible using these tiles.

## 2.3. Animals

Male ddY mice (Slc: ddY, six weeks old) were obtained from Japan SLC, Inc. (Shizuoka, Japan). All experiments were approved by the Experimental Animal Research Committee of Kyoto Pharmaceutical University and were performed according to the Guidelines for Animal Experimentation of Kyoto Pharmaceutical University.

# 3. Experimental

## 3.1. Manual radiolabelling

In order to prepare the $^{99m}$Tc-DTPA for scintigraphy, 10 μl DTPA saline solution and 10 μl sodium pertechnetate were added into a microtube and gently mixed with a pipette. After incubating the product for 5 min at room temperature, the chelating efficiency of $^{99m}$Tc-DTPA was verified by thin-layer chromatography (TLC), using acetone as the mobile phase. An autoradiograph of the reagents on the TLC sheet ($5 \times 1.5$ cm; TLC silica gel 60 $F_{254}$ aluminium plate; Merck, NJ, USA) was obtained using an Amersham Typhoon scanner (GE Healthcare, IL, USA). The chelation efficiency was estimated from the intensity of the autoradiograph using the equation:

$$\text{chelation efficiency} = \frac{(I_A - I_0)}{I_A + I_B - 2I_0} \times 100,$$

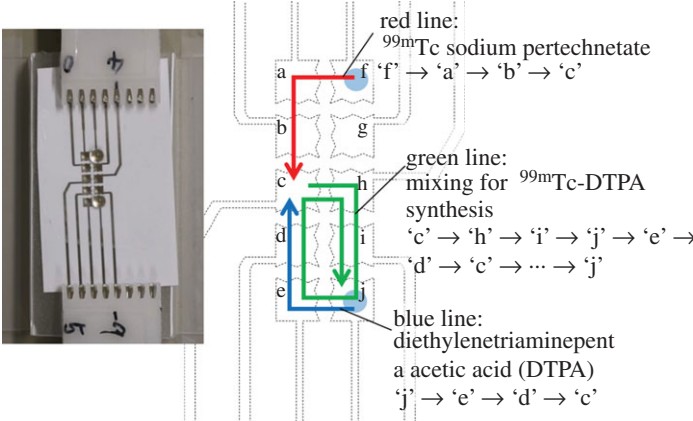

**Figure 4.** Droplet transport paths for the preparation of $^{99m}$Tc-DTPA. The tiles are designated by the letters 'a' to 'j'. The red line shows the path for DTPA, the blue line shows a second path for sodium pertechnetate, and the green line shows a third path for the DTPA/sodium pertechnetate mixture.

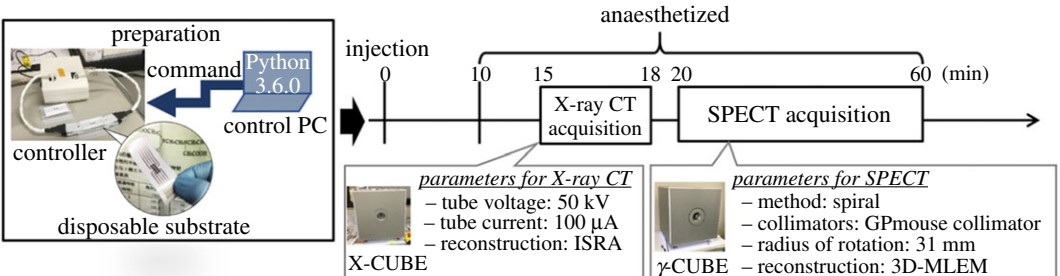

**Figure 5.** Schematic of the time flow and outline of the SPECT/CT imaging process. Fifteen minutes after injection of the radiolabelled reagents, X-ray computed tomographic images were acquired with the X-CUBE CT scanner, and SPECT images were subsequently acquired with the γ-CUBE SPECT (MOLECUBES, Ghent, Belgium) scanner. These processes were performed while the mice were anaesthetized.

where $I_A$ is integrated intensity of a $13 \times 13$ mm area where the prepared reagent was applied; $I_B$ is integrated intensity of a $13 \times 13$ mm area around unreacted $^{99m}$TcO$_4^-$ moved by capillary action and $I_0$ is integrated intensity of the background area on the TLC sheet. These values have been extracted using ImageJ.

## 3.2. Radiolabelling using the EWOD device

The process of combining and mixing two droplets was performed on the EWOD substrate tiles. The droplet transport path is shown in figure 4. An applied voltage of 300 V, switched at one-second intervals, was used to drive the droplets. At the beginning of the experiment, 10 µl sodium pertechnetate was placed on tile 'f', while 10 µl DTPA saline solution was placed on tile 'j' (figure 4). Subsequently, the sodium pertechnetate droplet was moved along the pathway from 'f' to 'c' (red line in figure 4), after which the DTPA droplet was moved along the path from 'j' to 'c' (blue line in figure 4). The two droplets were combined at 'c'. The merged droplet was circulated five times through a path used for solution mixing (green line in figure 4) and then deposited at 'j'. Five minutes after the mixing process, the reaction droplet was collected and the chelating efficiency of $^{99m}$Tc-DTPA was verified by TLC, using acetone as the mobile phase as in the case of the manual process.

## 3.3. SPECT/CT imaging in mouse

We demonstrated SPECT/CT imaging in mouse kidney using $^{99m}$Tc-DTPA prepared with the EWOD device.

SPECT/CT imaging was performed with γ-CUBE SPECT and X-CUBE CT scanners (MOLECUBES, Ghent, Belgium; figure 5). The mice were anaesthetized with isoflurane/O$_2$ (2.5% for induction, 2.0% for

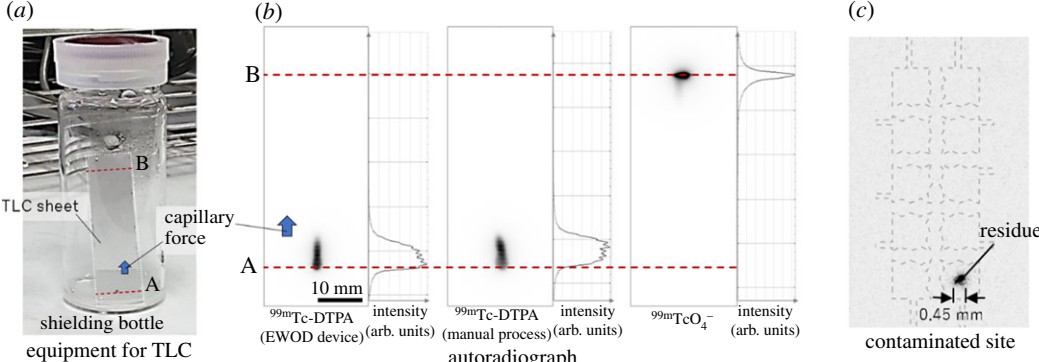

**Figure 6.** Autoradiography results. (*a*) Photo of the experimental equipment for TLC. The TLC sheet is soaked in acetone in a shielding bottle. Line 'A' shows a reagent drop point on the sheet, while line 'B' shows the position of $^{99m}TcO_4^-$ raised with acetone by capillary action. (*b*) Autoradiographs using EWOD-prepared $^{99m}Tc$-DTPA, manually prepared $^{99m}Tc$-DTPA and unreacted $^{99m}TcO_4^-$. The $^{99m}Tc$-DTPA remained at the origin 'A', while $^{99m}TcO_4^-$ was drawn up to 'B' by capillary action. (*c*) Contaminated site of the EWOD substrate. The residue remained as a spot (diameter, 0.45 mm) on the EWOD substrate.

maintenance), and their body temperature was kept constant with an integrated heating circuit. Subsequently, the mice were injected intravenously with the as-prepared $^{99m}Tc$-DTPA in saline solution (3.6 MBq/120 µl). Fifteen minutes after the intravenous administration of $^{99m}Tc$-DTPA, X-ray CT images were acquired for anatomic reference, and SPECT images were subsequently acquired.

CT images were acquired in helical scan mode with the following acquisition parameters: an X-ray source setting of 50 kVp/100 µA, 480 projections and a 1.75 spiral pitch, resulting in an acquisition time of 3 min. The CT projections were reconstructed using an image space reconstruction algorithm that yielded a voxel size of $0.1 \times 0.1 \times 0.1$ mm and image volume of $400 \times 400 \times 1017$.

SPECT images were subsequently acquired by helical orbit scanning with linear-stage motion and step-and-shoot camera motion. A multi-lofthole collimator (GPmouse, 48 loftholes) was mounted in the heptagonal SPECT system, and the following parameters were used for the SPECT scan: a $^{99m}Tc$ energy window (141 keV ± 10%), 473 projections, an 18° angle increment with a 1 mm bed step and an acquisition time of 40 min. The SPECT projection data were reconstructed using a three-dimensional maximum-likelihood-expectation maximization algorithm (four iterations) at an isotropic voxel size of 250 µm. Image analysis was performed using VivoQuant software (v. 4.0 patch1, inviCRO, LLC, Boston, MA, USA).

## 4. Results and discussion

The TLC results for the estimation of the chelation ratio are shown in figure 6. For scintigraphy, 1 µl of the prepared reagent was applied onto line 'A' of the TLC sheet, and one side of the sheet was soaked in acetone for 4–6 min (figure 6*a*). Figure 6*b* shows the autoradiographs of the EWOD-prepared $^{99m}Tc$-DTPA, manually prepared $^{99m}Tc$-DTPA and unreacted $^{99m}TcO_4^-$. Line 'B' in the figure shows the position of $^{99m}TcO_4^-$ raised by capillary action. The average chelation efficiency with EWOD processing, estimated from the intensity integration value, was 99.7 ± 0.13% (average of three trials), which was equivalent to or higher than that observed with manual processing of 99.4 ± 0.25% (average of four trials). This high efficiency indicates that the use of the EWOD device eliminates the need for additional processing to remove the unreacted substances. Further, to assess the contamination of the EWOD substrate, the substrate was inspected after completion of the reaction process (figure 6*c*). A spot of 0.45 mm in diameter was identified at position 'j' (figure 4). This spot represented the residue after the processed sample was collected with a pipette. Since the contaminated spot was small and predictable, the risk from radiation exposure during the replacement of the disposable substrates can be reduced. Additionally, the small field on the substrate for reagent manipulation is expected to minimize any spread of contamination.

Figure 7*a* shows the SPECT/CT images using the EWOD-prepared reagents, while figure 7*b* shows the results of the manually prepared reagents as the control. In figure 7*b*, only one kidney was visible because it was attributed to deterioration or missing function of the other kidney. The lack of the kidney image has no relation to the effect of the automation process discussed in this study, but we

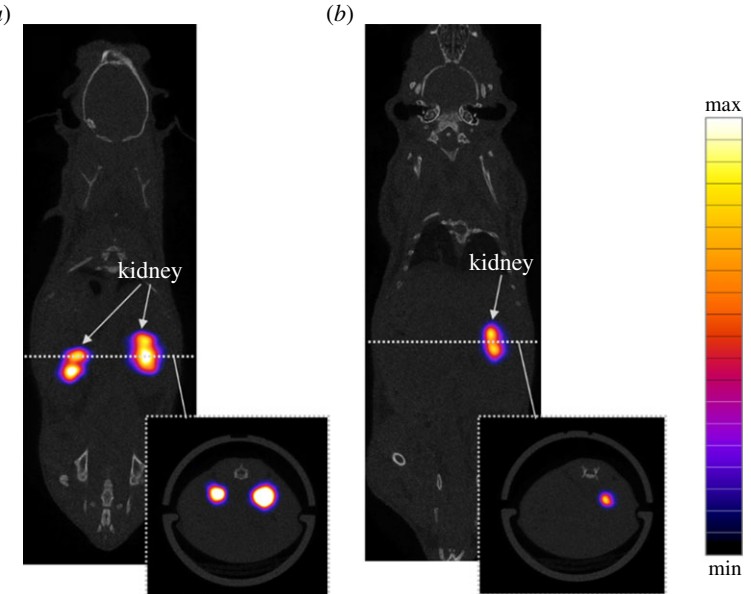

**Figure 7.** SPECT/CT image of mice injected with 3.6 MBq/120 µl $^{99m}$Tc-DTPA. The SPECT images in the blue-purple-red-yellow scale were merged with the CT images in grey scale (X-ray). (*a*) Coronal section image of a mouse injected with EWOD-processed $^{99m}$Tc-DTPA. (*b*) Coronal section image of a mouse injected with manually processed $^{99m}$Tc-DTPA. Owing to individual differences of the animal studied, only one kidney could be imaged in (*b*).

show the entire image to avoid intentional image editing. On the other hand, the other kidney images were well observed.

Using the EWOD-prepared reagents, we obtained high-resolution kidney images containing sufficient visualization information (e.g. high intensity and gradients) for the evaluation of mouse models for specific diseases. This result indicates that the manually preparing process of radiolabelled reagents can be replaced by the process using the proposed EWOD device as a human error reduction technique.

## 5. Conclusion

We have developed an open-style EWOD device for the preparation of radiolabelled reagents within a limited timeframe. This device can perform radiolabelling with simple operation and low running cost, using a cheaper and more compact system than those of conventional automatic equipment. The dimple structure mounted on the device minimized erroneous droplet movements caused by substrate tilt and distortion, and accurate microdroplet manipulation was realized. The thin-film substrate used in this EWOD device can be fabricated at low cost and is easily released from the device, making it suitable for disposable applications required in medical settings. The device performance was successfully demonstrated with an automatic chelation process using $^{99m}$Tc. We confirmed that $^{99m}$Tc-DTPA with a reaction efficiency of $99.7 \pm 0.13\%$ can be easily prepared. A SPECT imaging of small animal models proved that the EWOD device method can replace the manual labelling process because a reagent equivalent to that attained from the existing manual method could be prepared. Small amount manipulation by the EWOD device can be easily expanded to large-scale reagent preparation by parallel or continuous processing. Additionally, since the device is small and reagents can be easily taken in and out, it has a high affinity with the other existing technologies and can be combined with other bulky conveyance systems such as robotic arms. Thus, the proposed method can be used for evaluating small animals as well as diagnosing large animals including humans.

The high reaction efficiency obtained with our device is expected to eliminate any need for additional processing steps such as conventional unreacted substance removal and subsequent product concentration. Moreover, our proposed device facilitates rapid processing with high accuracy and reaction efficiency using a small and low-cost disposable film. Hence, it shows great potential for wide application as an essential tool for basic research as well as the development of new diagnostics using short-lived radioisotopes.

Ethics. All experiments were approved by the Experimental Animal Research Committee of Kyoto Pharmaceutical University and were performed according to the Guidelines for Animal Experimentation of the Kyoto Pharmaceutical University.

Data accessibility. TIFF data of autoradiography results including intensity information are submitted as supplementary files (electronic supplementary material). The average chelation efficiency in this article was estimated from the intensity integration value. Raw data have been uploaded to the Dryad Digital Repository: https://doi.org/10.5061/dryad.2z34tmpks [23].

Authors' contributions. Each author contributed mainly to conceptualization and methodology by K.M., H.K., T.I., S.A. and T.N..; validation by K.M., H.K. and Y.K.; writing—original draft preparation by K.M.; writing—review and editing by K.M. and T.I.; funding acquisition by K.M. and T.N. The authors meet all of the following criteria: (i) substantial contributions to conception and design, or acquisition of data, or analysis and interpretation of data; (ii) drafting the article or revising it critically for important intellectual content; (iii) final approval of the version to be published and (iv) agreement to be accountable for all aspects of the work in ensuring that questions related to the accuracy or integrity of any part of the work are appropriately investigated and resolved.

Competing interests. We have no competing interests. The funders had no role in the design of the study; in the collection, analyses or interpretation of data; in the writing of the manuscript, or in the decision to publish the results.

Funding. This work was partially supported by the Ministry of Education, Science, Sports and Culture, Grant-in-Aid for Scientific Research (C) (grant no. 20K05294) and Leading Initiative for Excellent Young Researchers (LEADER) in FY 2016 of the Ministry of Education, Culture, Sports, Science and Technology, Japan.

Acknowledgements. We would like to thank the 4 University Nano Micro Fabrication Consortium in Kawasaki, Japan (http://www.nano-micro.sakura.ne.jp/home/) who provided open facilities and experimental equipment for this work. We would like to thank Editage (www.editage.jp) for English language editing.

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
