## [Peer Review File · Royal Society Open Science]

Review History

RSOS-201809.R0 (Original submission)

Review form: Reviewer 1

Is the manuscript scientifically sound in its present form?

Yes

Are the interpretations and conclusions justified by the results?

Yes

Is the language acceptable?

Yes

Do you have any ethical concerns with this paper?

No

Have you any concerns about statistical analyses in this paper?

Yes

Recommendation?

Accept with minor revision (please list in comments)

Comments to the Author(s)

This work presented the microdroplet manipulating device base on the programmed electrowetting electrodes. The research findings can potentially be applied to future application in many microfluids controlling system.

Some points need to be clarified and revised:

- In the experimental section, the authors compare the manual process and EWOD device on TLC result and SPECT/CT imaging; however, there is no description on the manual process. Otherwise, the results could not be justified.
- The authors require to explain the data acquisition method and calculation of the average chelation efficiency of TLC result.
- How many times do the authors conduct the experiment on kidney images of reagent injection? The authors must provide more quantitative and statistical analysis to validate the performance of EWOD device vs the manual process.
- The real image of the device to present the dimple structure (which is one of the main features) on EWOD substrate will clarify the device design of this manuscript.
- A few minor typos, grammar hiccups should also be corrected, e.g. used in line 60, page 4.

Review form: Reviewer 2

Is the manuscript scientifically sound in its present form?

No

Are the interpretations and conclusions justified by the results?

Yes

Is the language acceptable?

Yes

Do you have any ethical concerns with this paper?

Yes

Have you any concerns about statistical analyses in this paper?

No

Recommendation?

Major revision is needed (please make suggestions in comments)

Comments to the Author(s)

General

The authors present in this paper a method for automatic radiolabelling of radiopharmaceuticals rapid and efficient that seems to have several advantages with regard to other procedures. The method is tested for ^{99m}Tc which has a relatively low half-life but should be studied for other lower half-lives radionuclides, so as to expand the utility of the method.

The paper is well written and structured and is a good contribution to the field. Nevertheless, it has some weaknesses that should be overcome before publication. Some of them are included in the following comments.

Specific

- 1.- The device is an evolution of a previous design. Perhaps some details on the improvements could help to properly judge the advances presented (and understand some of the sentences included).
- 2.- On the other hand, there is only a test with ^{99m}Tc . It could be convenient to test other radionuclides and labelling process so as to present a more general work. How difficult is to include in this paper another radionuclide?
- 3.- The experimental procedure is not detailed in depth since it seems to be related to previous works. This has to be done in order to complete the paper. For instance, the optimum depth of dimples seems to be some 52 micrometers. The figure must be justified.
- 4.- The radiolabelling strategy is crucial in this work. It seems to be very well established in the paper as shown in Fig. 3 with high efficiency (99.7%). How did the authors arrive to this merging strategy?. Was there a previous optimization work?
- 5.- Certainly Fig. 6 is difficult to understand. The absence of kidney image is attributed to kidney deterioration of the mouse. The explanation is not satisfactory at all, although it is not related (in principle) to the radiolabelling proposed method.

Decision letter (RSOS-201809.R0)

Dear Dr Mogi:

Title: Automatic radioisotope manipulation for small amount of nuclear medicine using an EWOD device with a dimple structure
Manuscript ID: RSOS-201809

Thank you for submitting the above manuscript to Royal Society Open Science. On behalf of the Editors and the Royal Society of Chemistry, I am pleased to inform you that your manuscript will be accepted for publication in Royal Society Open Science subject to minor revision in accordance with the referee suggestions. Please find the reviewers' comments at the end of this email.

The reviewers and handling editors have recommended publication, but also suggest some minor revisions to your manuscript. Therefore, I invite you to respond to the comments and revise your manuscript.

Because the schedule for publication is very tight, it is a condition of publication that you submit the revised version of your manuscript before 17-Jan-2021. Please note that the revision deadline will expire at 00.00am on this date. If you do not think you will be able to meet this date please let me know immediately.

Kind regards,
Dr Laura Smith
Publishing Editor, Journals

On behalf of the Subject Editor Professor Anthony Stace and the Associate Editor Dr Dattatray Late.

RSC Associate Editor:

Comments to the Author:

Authors presented interesting work on radioisotope manipulation for small amount of nuclear medicine using an EWOD device. Both the reviewer's have given good suggestions to revise the manuscript, authors can submit the revision as soon as possible.

RSC Subject Editor:

Comments to the Author:

(There are no comments.)

Reviewer comments to Author:

Reviewer: 1

Comments to the Author(s)

This work presented the microdroplet manipulating device base on the programmed electrowetting electrodes. The research findings can potentially be applied to future application in many microfluids controlling system.

Some points need to be clarified and revised:

- In the experimental section, the authors compare the manual process and EWOD device on TLC result and SPECT/CT imaging; however, there is no description on the manual process. Otherwise, the results could not be justified.
- The authors require to explain the data acquisition method and calculation of the average chelation efficiency of TLC result.
- How many times do the authors conduct the experiment on kidney images of reagent injection? The authors must provide more quantitative and statistical analysis to validate the performance of EWOD device vs the manual process.
- The real image of the device to present the dimple structure (which is one of the main features) on EWOD substrate will clarify the device design of this manuscript.
- A few minor typos, grammar hiccups should also be corrected, e.g. *usd* in line 60, page 4.

Reviewer: 2

Comments to the Author(s)

General

The authors present in this paper a method for automatic radiolabelling of radiopharmaceuticals rapid and efficient that seems to have several advantages with regard to other procedures. The method is tested for ^{99m}Tc which has a relatively low half-life but should be studied for other lower half-lives radionuclides, so as to expand the utility of the method.

The paper is well written and structured and is a good contribution to the field. Nevertheless, it has some weaknesses that should be overcome before publication. Some of them are included in the following comments.

Specific

- 1.- The device is an evolution of a previous design. Perhaps some details on the improvements could help to properly judge the advances presented (and understand some of the sentences included).

- 2.- On the other hand, there is only a test with ^{99m}Tc . It could be convenient to test other radionuclides and labelling process so as to present a more general work. How difficult is to include in this paper another radionuclide?.
- 3.- The experimental procedure is not detailed in depth since it seems to be related to previous works. This has to be done in order to complete the paper. For instance, the optimum depth of dimples seems to be some 52 micrometers. The figure must be justified.
- 4.- The radiolabelling strategy is crucial in this work. It seems to be very well established in the paper as shown in Fig. 3 with high efficiency (99.7%). How did the authors arrive to this merging strategy?. Was there a previous optimization work?.
- 5.- Certainly Fig. 6 is difficult to understand. The absence of kidney image is attributed to kidney deterioration of the mouse. The explanation is not satisfactory at all, although it is not related (in principle) to the radiolabelling proposed method.

Author's Response to Decision Letter for (RSOS-201809.R0)

See Appendices A & B.

RSOS-201809.R1 (Revision)

Review form: Reviewer 1

Is the manuscript scientifically sound in its present form?

Yes

Are the interpretations and conclusions justified by the results?

Yes

Is the language acceptable?

Yes

Do you have any ethical concerns with this paper?

No

Have you any concerns about statistical analyses in this paper?

No

Recommendation?

Accept as is

Comments to the Author(s)

The authors have addressed all my comments and therefore I support publication without further changes.

Review form: Reviewer 2

Is the manuscript scientifically sound in its present form?

Yes

Are the interpretations and conclusions justified by the results?

Yes

Is the language acceptable?

Yes

Do you have any ethical concerns with this paper?

No

Have you any concerns about statistical analyses in this paper?

No

Recommendation?

Accept as is

Comments to the Author(s)

Thank you for the effort in answering my questions and follow my suggestions. In my view the paper could be published in its actual form.

Decision letter (RSOS-201809.R1)

Dear Dr Mogi:

Title: Automatic radioisotope manipulation for small amount of nuclear medicine using an EWOD device with a dimple structure
Manuscript ID: RSOS-201809.R1

It is a pleasure to accept your manuscript in its current form for publication in Royal Society Open Science. The chemistry content of Royal Society Open Science is published in collaboration with the Royal Society of Chemistry.

On behalf of the Subject Editor Professor Anthony Stace and the Associate Editor Dr Dattatray Late.

RSC Associate Editor:
Comments to the Author:
Accept as is

RSC Subject Editor:
Comments to the Author:
(There are no comments.)

Reviewer(s)' Comments to Author:
Reviewer: 2

Comments to the Author(s)
Thank you for the effort in answering my questions and follow my suggestions. In my view the paper could be published in its actual form.

Reviewer: 1

Comments to the Author(s)
The authors have addressed all my comments and therefore I support publication without further changes.

Response to Reviewer 1 Comments

December 16th, 2021

From: Katsuo Mogi

Title: Automatic radioisotope manipulation for small amount of nuclear medicine using an EWOD device with a dimple structure
Manuscript ID: RSOS-201809

Dear Sir,

We thank you for careful reading our manuscript and for giving accurate instructions. Your valuable comments helped me very much to improve the quality as well as the readability of the paper. On the basis of your comments, I added new description and figures. I hope the revision could meet the conditions to be considered as the publication.

- In the experimental section, the authors compare the manual process and EWOD device on TLC result and SPECT/CT imaging; however, there is no description on the manual process. Otherwise, the results could not be justified.

Ans.> Thank you for your comment. It is very important point as your mention. So, we have added the following new section to the Experiment chapter, and also made related corrections in the chapter.

[4. Experimental, first paragraph]

Manual radiolabelling

In order to prepare the ^{99m}Tc -DTPA for scintigraphy, 10 μL DTPA saline solution and 10 μL sodium pertechnetate were added into a microtube and gently mixed with pipette. After incubating the product for 5 minutes at room temperature, the chelating efficiency of ^{99m}Tc -DTPA was verified by thin-layer chromatography (TLC), utilising acetone as the mobile phase. An autoradiograph of the reagents on the TLC sheet (5×1.5 cm; TLC silica gel 60 F₂₅₄ aluminium plate; Merck, NJ, USA) was obtained using an Amersham Typhoon scanner (GE Healthcare, IL, USA).

- The authors require to explain the data acquisition method and calculation of the average chelation efficiency of TLC result.

Ans.> thank you for the beneficial comment. As you mentioned, I added new sentences for explanation of TLC data as follow.

[4. Experimental, Manual radiolabelling, I6-I12]

The chelation efficiency was estimated from the intensity of the autoradiograph using the equation:

$$\text{Chelation efficiency} = \frac{(I_A - I_0)}{I_A + I_B - 2I_0} \times 100$$

Where I_A is integrated intensity of a 13mm x 13mm area where the prepared reagent was applied; I_B is integrated intensity of a 13mm x 13mm area around unreacted $^{99m}\text{TcO}_4^-$ moved by capillary action; and I_0 is integrated intensity of the background area on the TLC sheet. These values have been extracted using the ImageJ.

- How many times do the authors conduct the experiment on kidney images of reagent injection? The authors must provide more quantitative and statistical analysis to validate the performance of EWOD device vs the manual process.

Ans.> We conducted one time of animal experiments for kidney scintigraphy using reagent obtained by manual preparation. And we conducted two times of that by the EWOD device. The number of this experiments is not large.

However, We have important bioethical barriers worldwide. Given the global importance of bioethics, it is necessary to reduce the number of times animals are used in the basic evaluation of engineering instruments.

So, we conducted kidney scintigraphy with ^{99m}Tc -DTPA as the most standard and simple experiment system for the device demonstration. This scintigraphy has been also used clinically for a long time. The purpose of this scintigraphy is to obtain images of the kidneys, so no quantitative evaluation of any metabolites or biotransforms is necessary. Our experiments provide clear images of the kidneys, which is sufficient for the purpose of demonstrating this device usability.

We must ethically reduce the number of animal experiments, but in the future we will continue to conduct animal experiments that specialize in specific diseases and diagnoses.

I am very sorry, but if it doesn't seem to be the answer you want, could you please comment again?

- The real image of the device to present the dimple structure (which is one of the main features) on EWOD substrate will clarify the device design of this manuscript.

Ans.> Thank you for your comment. We embedded the real image of the substrate and the dimple structure in Figure 2 as follows. Additionally, new figure 3 and sentences, and a formula for dimple-shape estimation was add in the article for explanation of the dimple designing.

Figure 2. Schematics of the EWOD-substrate structure. (a) Outline and electrode pattern of the substrate; 2 mm square jagged tiles are aligned with 300 μm spacing. (b) Cross sectional view of line A-A'. The thin-film substrate is composed of an insulating membrane and the paper on which the electrodes are printed. A dimple structure with a spherical-cap shape is formed on each tile. (c) Fabrication process of a dimple structure on the EWOD substrate.

Figure 3. Design index and optimum shape of the Dimple structure. (a) The dimple structure is designed using the coefficient (adhesion energy: K) obtained by Wolfram's empirical formula as an index. The K_{flat} is the coefficient of a flat substrate. The optimum shape of dimple structure which has the best balance between mobility and stability is achieved with $K = 1.3 K_{flat}$. (b) Contour plot of a dimple structure. Point B shows the centre of the dimple structure, and point B' shows a flat position. Depth of the dimple structure is $52.9 \pm 2.5 \mu\text{m}$, which is the average and the standard deviation of five dimples.

[3. Materials and Methods, EWOD device with dimple str., 3rd paragraph, 13-18]
 The dimple structure can increase the stability of the droplets on each tile and avoid the perturbation that was a major challenge for EWOD. On the other hand, it was necessary to design the optimum shape of the structure because excessive stability causes droplet sticking. We have already shown a method for estimating the optimum dimple structure from the coefficient (the adhesion energy: K) obtained by Wolfram's experimental formula (1) ²².

$$K = \frac{mg \sin \alpha}{2\pi r}, \quad (1)$$

where m is the droplet mass, g is the acceleration due to gravity, α is the critical tilting angle, and r is the droplet contact radius. When the K is approximately 1.3 times that of a flat substrate, the dimples have the best balance between mobility and stability (Figure 3a). Figure 3b shows the optimum dimple structure with depth of $52.9 \pm 2.5 \mu\text{m}$ used in this study.

- A few minor typos, grammar hiccups should also be corrected, e.g. used in line 60, page 4.

Ans. > I am sorry for the minor typos. I checked the text again. Also, a proofing company checked the text, so I attached the calibration certificate from the company.

Appendix B

Response to Reviewer 2 Comments

December 17th, 2021

From: Katsuo Mogi

Title: Automatic radioisotope manipulation for small amount of nuclear medicine using an EWOD device with a dimple structure
Manuscript ID: RSOS-201809

Dear Sir,

We thank you for careful reading our manuscript and for giving accurate instructions. Your valuable comments helped me very much to improve the quality as well as the readability of the paper. On the basis of your comments, I added new description and figures. I hope the revision could meet the conditions to be considered as the publication.

1.- The device is an evolution of a previous design. Perhaps some details on the improvements could help to properly judge the advances presented (and understand some of the sentences included).

Ans. > Thank you for your beneficial comment. We strongly agree with you. Due to some unfriendly omissions for readers who are unaware of our previous article, we have added an outline of device fabrication in Figure 2. That is include the evolution of our previous designing. The number of electrode tiles in this article is 10. It was also evolution of previous design of 6 tiles. 10 tiles make it easy to operate not only one but also two droplets. But, to avoid confusion for readers who do not know our previous article, I have only described the design of the 10 tiles.

Figure 2. Schematics of the EWOD-substrate structure. (a) Outline and electrode pattern of the substrate; 2 mm square jagged tiles are aligned with 300 μm spacing. (b) Cross sectional view of line A-A'. The thin-film substrate is composed of an insulating membrane and the paper on which the electrodes are printed. A dimple structure with a spherical-cap shape is formed on each tile. (c) Fabrication process of a dimple structure on the EWOD substrate.

2.- On the other hand, there is only a test with 99mTc. It could be convenient to test other radionuclides and labelling process so as to present a more general work. How difficult is to include in this paper another radionuclide?.

Ans. > As your point, it is an experiment with 99mTc only, but it is very meaningful demonstration because the 99mTc has been used clinically for a long time. Hence, we believe that the Tc experiment is sufficient for this purpose of demonstrating the usefulness of the device. On the other hand, as you pointed out, we are also interested in the use of various radioisotopes. However, in order to handle other radioisotopes, it takes a lot of time to prepare for the experiment in consideration of half-life and safety, so it is difficult to include in this study and we will experiment in our next study. In our next study, we will demonstrate the usability of the device by manipulating diagnostic nuclides (111In, 67 / 68Ga, etc.) and therapeutic nuclides (90Y, 177Lu, etc.) to label peptides and antibodies.

3.- The experimental procedure is not detailed in depth since it seems to be related to previous works. This has to be done in order to complete the paper. For instance, the optimum depth of dimples seems to be some 52 micrometers. The figure must be justified.

Ans. > Thank you for your beneficial comment. We apologize for these unfriendly omissions for readers who are unaware of our previous article. The article has been revised, including the answer to Q1. And we added sentences and equation, and new Figure 3 to explain the dimple design as follows.

[3. Materials and Methods, EWOD device with dimple str., 3rd paragraph, 13-18]

We have already shown a method for estimating the optimum dimple structure from the coefficient (the adhesion energy: K) obtained by Wolfram's experimental formula (1) ²².

$$K = \frac{mg \sin \alpha}{2\pi r}, \tag{1}$$

where m is the droplet mass, g is the acceleration due to gravity, α is the critical tilting angle, and r is the droplet contact radius. When the K is approximately 1.3 times that of a flat substrate, the dimples have the best balance between mobility and stability (Figure 3a). Figure 3b shows the optimum dimple structure with depth of $52.9 \pm 2.5 \mu\text{m}$ used in this study.

Figure 3. Design index and optimum shape of the Dimple structure. (a) The dimple structure is designed using the coefficient (adhesion energy: K) obtained by Wolfram's empirical formula as an index. The K_{flat} is the coefficient of a flat substrate. The optimum shape of dimple structure which has the best balance between mobility and stability is achieved with $K = 1.3 K_{flat}$. (b) Contour plot of a dimple structure. Point B shows the centre of the dimple structure, and point B' shows a flat position. Depth of the dimple structure is $52.9 \pm 2.5 \mu\text{m}$, which is the average and the standard deviation of five dimples.

4.- The radiolabelling strategy is crucial in this work. It seems to be very well established in the paper as shown in Fig. 3 with high efficiency (99.7%). How did the authors arrive to this merging strategy? Was there a previous optimization work?

Ans. > ^{99m}Tc -DTPA has been used clinically for a long time and its preparation has been optimized. In this study, the amount of precursor and reaction time were optimized based on the general preparation protocol. We also examined the conditions for the number of mixings on the device. So, this merging strategy was highly efficient, so we considered this strategy to be sufficient.

5.- Certainly Fig. 6 is difficult to understand. The absence of kidney image is attributed to kidney deterioration of the mouse. The explanation is not satisfactory at all, although it is not related (in principle) to the radiolabelling proposed method.

Ans. > Thank you for your helpful advice. I added a sentence as follows. Since the mouse without kidneys was individual injected with the manually prepared reagent, it was judged that there was no problem with the purpose of this experiment. I would appreciate your understanding.

Actually, mice lacking kidneys are not uncommon. From a bioethical point of view, we do not want to carry out unnecessary animal experiments. However, if this answer is not enough for your question, let us know. We can do additional animal experiment.

[5 Results and Discussion, 2nd paragraph, 12-15]

In the Figure 7B, only one kidney was visible because it was attributed to deterioration or missing function of the other kidney. The lack of the kidney image has no relation to the effect of the automation process discussed in this study, **but we show the entire image to avoid intentional image editing.**